# Enhancement of Dendrobine Production by CRISPR/Act3.0-Mediated Transcriptional Activation of Multiple Endogenous Genes in *Dendrobium* Plants

**DOI:** 10.3390/ijms26041487

**Published:** 2025-02-11

**Authors:** Meili Zhao, Zhenyu Yang, Jian Li, Feng Ming, Demin Kong, Haifeng Xu, Yu Wang, Peng Chen, Xiaojuan Duan, Meina Wang, Zhicai Wang

**Affiliations:** 1Shenzhen Key Laboratory for Orchid Conservation and Utilization, the National Orchid Conservation Center of China and the Orchid Conservation & Research Center of Shenzhen, Shenzhen 518114, China; 17720798949@163.com (M.Z.); yangzhenyu.eco@foxmail.com (Z.Y.); damykong@126.com (D.K.); h.f.xu@163.com (H.X.); 15361439572@163.com (Y.W.); comeondxj@126.com (X.D.); 2Key Laboratory of National Forestry and Grassland Administration for Orchid Conservation and Utilization, the National Orchid Conservation Center of China and the Orchid Conservation & Research Center of Shenzhen, Shenzhen 518114, China; 3Shanghai Key Laboratory of Plant Molecular Sciences, College of Life Sciences, Shanghai Normal University, Shanghai 200234, China; fming@fudan.edu.cn

**Keywords:** *Dendrobium catenatum*, *Dendrobium nobile*, dendrobine, multigene stacking, CRISPR/Act3.0

## Abstract

Dendrobine, a significant medicinal compound, typically accumulates at low concentrations within several *Dendrobium* species, including *Dendrobium nobile*, *Dendrobium catenatum*, and *Dendrobium moniliforme*. As *D. nobile* and *D. catenatum* are established ingredients in traditional Chinese medicine and have been cultivated extensively, they present ideal plant chassis for upscaling dendrobine production for industrial and research applications. This study employed two approaches: the ectopic overexpression of seven genes through multigene stacking and the activation of multiple key endogenous genes in the dendrobine synthesis pathway using CRISPR/Act3.0 in either *D. nobile* or *D. catenatum*. These methods enhanced dendrobine production in transiently infiltrated leaves by 30.1% and transgenic plants by 35.6%. The study is the first to apply CRISPR/Act3.0 to *Dendrobium* orchids, enhancing dendrobine production, and thus laying a solid foundation for further improvements. CRISPR/Act3.0 is a recently developed technique that demonstrates high efficiency in model plant species, including rice, maize, and *Arabidopsis*. By combining CRISPR with transcriptional regulatory modules, activation of multiple endogenous genes in the metabolic pathway can be achieved. The successful application in orchid molecular breeding reveals promising potential for further exploration.

## 1. Introduction

Orchid species, such as *Bletilla striata*, *Gastrodia elata*, and *Dendrobium* spp., are highly valued for their medicinal and ornamental properties. However, due to stringent habitat requirements, climate change, and human activities, their populations are continuously declining, with many species now critically endangered [1]. Furthermore, *Dendrobium* plants cultivated under greenhouse conditions often exhibit lower and inconsistent levels of active compounds compared to their wild counterparts, influenced by factors such as growth media and hormone treatments [2]. *Dendrobium* plants contain a diverse array of pharmacologically active components, including over 35 different compounds such as alkaloids, polysaccharides, flavonoids, phenanthrenes, bibenzyls, sesquiterpenoids, and coumarins [3]. These components demonstrate significant anticancer, antioxidant, and immunomodulatory activities [4]. Currently, the majority of alkaloids isolated from *Dendrobium* species are sesquiterpene alkaloids [5]. These compounds are essential medicinal components in *Dendrobium* plants, and their content is often utilized as a control standard for quality evaluation [6]. Dendrobine, a major sesquiterpene alkaloid with high medicinal and commercial value, is typically present in low concentrations in several *Dendrobium* species [7]. It exhibits analgesic and antipyretic effects, can reduce heart rate and blood pressure, slow respiration, induce moderate hyperglycemia, and alleviate symptoms of barbiturate poisoning [8]. The major source of dendrobine to date is still extraction from *D. nobile*, which is insufficient due to the long growth period and low concentration. Therefore, improving the production of dendrobine in *Dendrobium* plants and diversifying the source materials is of great significance for further development and utilization.

Dendrobine synthesis primarily occurs through the methylerythritol phosphate (MEP) and the mevalonate (MVA) pathways [4,9]. Transcriptome sequencing, gene ontology, and Kyoto Encyclopedia of Genes and Genomes analyses have identified key genes in the synthetic pathway [10], providing opportunities to enhance dendrobine production. The advancements in metabolic engineering through CRISPR-mediated transcriptional activation have been extensively reviewed previously [11]. The CRISPR/Cas9 gene editing and transcriptional regulation technology, known for its high efficiency and precision, has demonstrated significant potential in plant gene functional studies [12]. Notably, CRISPR/Act3.0 has emerged as a new generation transcriptional activation system, enabling the transcriptional activation of multiple genes through the designing and tandem expression of various single-guide RNAs (sgRNAs) [13]. This system, comprising dCas9-VP64, gR2.0 scaffold, 10×GCN4 SunTag, and the newly developed 2×TAD activator [14], has exhibited strong transcriptional activation capabilities for endogenous genes in *Arabidopsis* and rice [15,16], although its application in orchids remains unreported.

Our previous study constituted seven dendrobine biosynthesis-related genes and generated transgenic plants in *D. catenatum* [17]. In this investigation, we further assessed gene expression and dendrobine content in these transgenic plants. Furthermore, we applied CRISPR/Act3.0 to achieve multiple endogenous gene activations in the dendrobine synthesis pathway of *D. nobile* and *D. catenatum*. Additionally, utilizing the in planta transformation method we previously developed for *D. catenatum* [18], we generated CRISPR/Act3.0 transgenic plants. This study applies newly developed technologies to perennial and transformation-recalcitrant *Dendrobium* orchids, establishing a foundation for molecular breeding in these species.

## 2. Results

### 2.1. Overexpression of Multiple Stacked Genes Enhanced Dendrobine Production in Transgenic D. catenatum

Currently, the genetic manipulation of orchids is challenging due to their long growth periods and recalcitrance to regeneration. Although there are plenty of studies using orchid materials, research involving transgenic orchids is sporadic. To enhance dendrobine production in *D. catenatum*, we previously stacked multiple genes (SG, seven genes) involved in the dendrobine synthesis pathway into one vector (Figure 1A) and transformed it into *D. catenatum*. Several transgenic seedlings were obtained with activated farnesyl pyrophosphate synthase (FPPS) expression, indicating enhanced dendrobine production [17]. As a continuation of this research, we monitored the growth of the transgenic plants and measured the dendrobine content in the plants using LC/MS (liquid chromatography–mass spectrometry), which was previously unattainable due to material constraints. The two-year-old SG-multigene transgenic plants displayed normal growth and exhibited phenotypes comparable to those of the empty vector control transformed plants (Figure 1B). Upregulation of each individual gene, with the exception of METTL23 (methyltransferase-like protein 23), was observed at varying levels, suggesting that the transformed multigene construct is functional in vivo. Given the use of repeated *Prrn* promoters (promoter of tobacco-derived 16S rRNA) and *Trbcl* terminators (terminator of tobacco-derived ribulose-1,5-bisphosphate carboxylase/oxygenase large subunit) in the construct, it is possible that gene silencing of transgenes occurred, potentially explaining the low expression detected for *METTL23* (Figure 1C). Furthermore, the expression of *FPPS* was upregulated in the multigene transgenic plants (Figure 1D), indicating activation of the dendrobine synthesis pathway. To quantify dendrobine content in the transgenic plants, LC/MS analysis revealed a significant increase from 34.6 µg/g (fresh weight) in the empty vector (EV) control plants to 46.4 µg/g (FW) in the multigene transgenic plants (Figure 1E and Appendix A). It is worth noting that the LC/MS measurement of dendrobine provides more precise results compared to the high-performance liquid chromatography (HPLC) method employed in previous studies [17], which offered a relative comparison and quantification.

### 2.2. Transcriptional Activation of MCT (2-C-Methyl-d-Erythritol 4-Phosphate Cytidylyltransferase) by CRISPR/Act3.0 Enhanced Dendrobine Production in D. catenatum Leaves

The overexpression of multiple stacked transgenes in transgenic *D. catenatum* resulted in a statistically significant, albeit modest, increase in dendrobine production. Consequently, we explored the use of CRISPR/Act3.0, a potent transcriptional regulator, to more effectively upregulate endogenous genes and enhance dendrobine synthesis. We designed and compared five sgRNAs targeting *MCT*, a key gene previously identified in the dendrobine synthesis pathway [17] (Figure 2A). These sgRNAs were designed and evaluated using the CRISPOR online tool (http://crispor.gi.ucsc.edu/, accessed on 1 March 2022). Because in many cases the transcription starting site (TSS) is hard to precisely predict, the candidate sgRNAs were designed within 200 base pairs upstream of the initiation codon ATG. Based on comprehensive scoring results, MCT-sgRNA3 received the highest score (Figure 2B). Additional scores for each sgRNA were provided by CRISPR RGEN and inDelphi as well. To validate the efficacy of these sgRNAs, we conducted a dual-luciferase reporter assay in *Dendrobium* protoplasts approximately 12 h after the co-transformation of CRISPR-Ac3.0-sgRNA and REN/LUC reporter plasmids. MCT-sgRNA3 exhibited the strongest luciferase (LUC) signal, while MCT-sgRNA2 and MCT-sgRNA5 showed comparatively lower effectiveness (Figure 2C). Subsequently, we selected MCT-sgRNA3 and MCT-sgRNA2 for further testing in transiently infiltrated *Dendrobium* leaves. The experimental data indicated that MCT-sgRNA3 not only outperformed MCT-sgRNA2 in activating *MCT* expression (measured 6 and 24 h post transformation) but also significantly increased dendrobine content (measured 5 days post transformation), consistent with the CRISPOR scores (Figure 2D,E). These results demonstrated that CRISPOR, typically used to predict sgRNA binding and editing efficiencies, can also predict binding and activation efficiency. Notably, the increase in dendrobine content achieved through targeting a single gene in partially transformed leaves was comparable to that obtained through SG-multigene overexpression. This suggests that CRISPR/Act3.0 exhibits greater potential for enhancing dendrobine production.

### 2.3. Transcriptional Activation of Multiple Endogenous Genes by CRISPR/Act3.0 Enhanced Dendrobine Production in D. nobile Leaves

It has been well established that *D*. *nobile* possesses a highly activated synthesis pathway and the highest level of dendrobine production among *Dendrobium* species [19,20]. It is conceivable that the further activation of key genes in this synthetic pathway could result in a more substantial increase in dendrobine production. To investigate this, we transiently overexpressed CMEAO (copper methylamine oxidase), a gene previously identified as a positive regulator for dendrobine synthesis [17], in leaves of *D. nobile* (Figure 3A). As anticipated, the overexpression of *CMEAO* elevated dendrobine levels in *D. nobile* in comparison to empty vector infiltrated control, suggesting that the function of the *CMEAO* gene is conserved across *Dendrobium* species (Figure 3B). Additionally, our previous research demonstrated that the overexpression of several *BGLU* (*β-Glucosidase*) genes enhanced alkaloid synthesis [21]. Given that dendrobine is a sesquiterpene alkaloid, it is of interest to determine whether these genes play functional roles in dendrobine synthesis. The results indicated that the transient overexpression of *BGLU2*, *6*, and *8* increased dendrobine content, while *BGLU13* may function as a negative regulator for dendrobine synthesis (Appendix A) in *D. catenatum*. Subsequently, to further enhance dendrobine production in *D. nobile*, we employed the multiple sgRNA construction strategy in conjunction with CRISPR/Act3.0 activation domains. This approach enabled us to target eight endogenous genes including *CMEAO*, *BGLU2* and *BGLU8*, in the dendrobine synthesis pathway using a single construct (Figure 3C). The results demonstrated that the transient activation of multiple endogenous genes not only significantly increased the expression levels of individual genes but also led to a marked increase in the expression of *FPPS*, which serves as an activation marker for dendrobine synthesis (Figure 3D,E). Despite the high efficiency of the transient expression of CRISPR/Act3.0 in the leaves, we observed only a modest increase in dendrobine synthesis (Figure 3F). This may be attributed to the dendrobine synthesis pathway already been highly activated; thus, further enhancing the expression of key genes may not result in a significant increase in dendrobine production. However, this system might prove effective in *D. catenatum*, which exhibits much lower levels of dendrobine content.

### 2.4. Transcriptional Activation of Multiple Endogenous Genes by CRISPR/Act3.0 Improved Dendrobine Production in Transgenic D. catenatum Plants

Subsequent activation of multiple endogenous genes in *D. nobile* yielded only a marginal increase in dendrobine production. As a result, we shifted our focus to other *Dendrobium* species with lower dendrobine content, such as *D. catenatum*, to further assess the potential of this activator construct. We targeted the promoters of four key genes, including *MCT*, *STR1* (*Strictosidine Synthase 1*), *CYP94C1* (*Cytochrome P450 94C1*), and *HMGR* (*3-Hydroxy-3-Methylglutaryl Coenzyme-A Reductase*), using CRISPR/Act3.0, with two sgRNAs for each gene (eight sgRNAs in total, collectively designated as 8sgRNA) (Figure 4A). To evaluate the efficiency of the sgRNAs, we transiently expressed the CRISPR/Act3.0-8sgRNA vector in protoplast cells. The results indicated upregulation of all four targeted genes, with *MCT* showing an increase of more than six-fold (Figure 4B). Subsequently, we transiently expressed the activation construct in *D. catenatum* leaves to further assess its contribution to dendrobine synthesis. The results demonstrated that the endogenous activation of the four targeted genes by CRISPR/Act3.0 (Figure 4C) enhanced dendrobine production (Figure 4D), highlighting the significant potential of the activator for improving dendrobine synthesis. Encouraged by these findings, we generated transgenic *D. catenatum* plants containing the CRISPR/Act3.0-8sgRNA construct through in planta transformation [18]. Three positive transgenic plants for the CRISPR/Act3.0-8sgRNA and two for the empty vector control were obtained (Figure 4E,F), as confirmed by genomic PCR amplification of the co-transformed hygromycin resistance gene *hptII* (Figure 4G and Appendix A). To mitigate the influence of untransformed plants, the isolated positive individuals for both the 8sgRNA and vector control were combined, respectively, and replanted into new bark-pots (Figure 4H). While no visual differences were apparent, the analysis of the targeted gene expression revealed a significant increase in *STR1*, whereas the other three genes did not exhibit similar upregulation (Figure 4I), suggesting potential gene silencing. The downregulation of the other two genes compromises the contribution of *STR1* to dendrobine synthesis. Consistent with this finding, a slight upregulation of the *FPPS* gene was also observed (Figure 4J), which was accompanied by suboptimal improvements in dendrobine production (Figure 4K).

## 3. Discussion

The study applied CRISPR/Act3.0 for the first time to probe the potential of this new technology in *Dendrobium* species. Endogenous activation of dendrobine synthesis genes by CRISPR/Act3.0 resulted in enhanced dendrobine production, with more profound effects observed in *D. catenatum*. Through expression manipulation of multiple genes in the synthesis pathway, an increase of up to 35.6% in dendrobine production was achieved. This will contribute to increasing the added value of *D. catenatum* and lay a solid foundation for further improvements. The major challenge, however, is still the tedious work of genetic transformation and the long duration of obtaining transformants. So far, the most common method for the genetic transformation of *Dendrobium* species is mediated by *Agrobacterium* infiltration of protocorm. Yet, no standard guidelines are available. More than ten rounds of antibiotic selection are needed to obtain transformants. To improve transformation efficiency and accelerate the process, in planta transformation was developed previously and utilized in this study to create stable transgenic *D. catenatum* plants. Because no antibiotic selection was applied in the early stages of transformation, chimerism might occur. Further evaluation of the chimerism and the physiological changes in the transgenic plants is needed.

Given the necessity of employing multiple promoters for multigene expression in a single vector, and the current lack of effective promoters, researchers often reuse a particular promoter in the construct [22]. In our SG-multigene transgenic *Dendrobium* plants, we observed a downregulation of *METTL23*. This downregulation may be attributed to the repeated use of *Prrn* promoters and *Trbcl* terminators in the constructs, as well as the potential impacts of RNA interference and positional effects. Further investigation is needed to clarify these factors. It should be noted that the *Prrn* promoter and *Trbcl* terminator are chloroplast elements. However, we could not obtain the chloroplast transformation vector when we started the project. Later, we learned that the chloroplast elements can be expressed in the nucleus to some extent [23]. As a result, we decided to perform nuclear transformation using these elements and detect efficient expression of the target genes. While most of the target genes in the construct are predicted to function in dendrobine synthesis, experimental verification of each individual gene remains necessary. Notably, variations in dendrobine content were observed among different batches of empty vector control transformed *D. catenatum* or *D. nobile* materials. The dendrobine content in the greenhouse-grown *D. catenatum* plants used for transient infiltration of leaves ranged from 1.5 µg/g DW (Figure 2E) to 11.5 µg/g DW (Figure 4D), as determined by more precise LC/MS analysis. However, the relative quantification of dendrobine by HPLC reaches approximately 3000 µg/g DW. This study employs HPLC to compare the relative quantification of dendrobine and to screen candidate genes, while LC/MS is used for the precise determination of dendrobine content. Moreover, *D. catenatum* plants grown in the culture room accumulated higher levels of dendrobine compared to those grown in the greenhouse, ranging from 35 µg/g FW (Figure 1E) to 70 µg/g FW (Figure 4K). Previous studies have shown that the total alkaloid accumulation of *D. nobile* decreases with increasing growth years [24,25]. Consistent with these findings, the absolute quantity of dendrobine also decreased over time [26]. Considering the extended experimental period, the selection of different batches of materials at various growth stages may contribute to this observed variation. Nevertheless, the overall enhancement tendency of CRISPR-Act3.0 on endogenous gene expression and thus boosting dendrobine production remains consistent.

CRISPOR, a genome-based algorithmic tool for designing sgRNAs, integrates multiple factors such as targeting, specificity, and potential side-effects, and provides comprehensive analyses of the design results, making it widely used in research [27]. In this study, utilizing CRISPOR, the highest-scoring MCT-sgRNA3 and the low-scoring MCT-sgRNA2 were selected and evaluated for functionality in *Dendrobium* leaves. The experimental data demonstrated that MCT-sgRNA3 not only surpassed MCT-sgRNA2 at the gene expression level but also significantly increased the content of dendrobine, a result highly consistent with the CRISPOR score. According to the protocol, protoplasts were cultured for 12 h before gene expression analysis. The activation of sgRNA2, sgRNA3, and sgRNA5 was detected, with sgRNA3 exhibiting the highest activity. Subsequently, sgRNA3 was further tested in transiently infiltrated leaves, with sgRNA2 serving as the control. The activation of *MCT* expression was analyzed at 6 and 24 h, while dendrobine content was measured five days after infiltration. The activation of *MCT* expression was assessed at different time points using both the protoplast and transient leaf expression systems. It is evident that the final concentration of dendrobine results from the cumulative expression of *MCT* during the five-day culture, and sgRNA2 was consistently less effective than sgRNA3. Therefore, sgRNAs with the highest scores are recommended. Additionally, a dual luciferase reporter assay was conducted for MCT-sgRNA1 to MCT-sgRNA5 in the *Dendrobium* leaf protoplast system. The results indicated that MCT-sgRNA3, when co-transformed with CRISPR/Act3.0, exhibited the highest transcriptional activation, further validating the accuracy of CRISPOR prediction. CRISPRa efficiency is sensitive to the sgRNA targeting site relative to the transcription start site (TSS), and effective sgRNAs are often located within a 200 bp region upstream of the TSS. However, the precise annotation of the TSS in the promoter is still challenging due to the complex architecture of promoter sequences [28]. For this reason, we designed sgRNAs for CRISPR-Act3.0 targeting the proximal promoter between -200 bp and the initiation codon ATG, then evaluated using CRISPOR, verified using protoplasts, or designed multiple sgRNAs for one specific gene.

*CMEAO*, a recently identified gene encoding copper methylamine oxidase, is predicted to function in the biosynthesis of tropane, piperidine, and isoquinoline alkaloids [17]. Previous research demonstrated that overexpression of *CMEAO* in *D. catenatum* enhanced dendrobine production [17]. In this study, overexpression of *CMEAO* in *D. nobile* similarly resulted in a significant increase in dendrobine content, suggesting a conserved function of *CMEAO* as a positive regulator in the dendrobine synthesis pathway. Utilizing CRISPR/Act3.0 technology, eight candidate genes in *D. nobile* underwent multi-gene activation. While the expression level of each gene increased, the enhancement of dendrobine content was notable but not substantial. This limited enhancement may be attributed to *D. nobile* already possessing the highest dendrobine content among *Dendrobium* species [29], making further increases more challenging from this elevated baseline. Additionally, these eight activated genes are situated at different nodes of the metabolic pathway, which has not been fully elucidated. The exclusion of potentially crucial enzymes or genes from the construction might have restricted further enhancement of dendrobine content. Even though the whole transcription profile has not been checked, the increased production of dendrobine is thought to result from the upregulation of multiple positive genes in the synthesis pathway. *FPPS* encodes a key enzyme responsible for the precursor synthesis of dendrobine. Overexpression of *FPPS* is corresponding to enhanced dendrobine production [30]. Therefore, the upregulation of *FPPS* indicates the activation of the synthesis pathway.

Plant genetic transformation is a sophisticated process involving the introduction of target genes into plant cells through physical, chemical, or biological methods, followed by the selection and cultivation of transgenic plants [31]. Moreover, researchers frequently encounter challenges such as limitations in gene transfer technique, restrictions in the genotype of explants or species, random integration of exogenous DNA, difficulties in regeneration, and inefficiency of transformation [32]. In this study, we generated CRISPR/Act3.0 transgenic *D. catenatum* plants using in planta transformation technique [18]. However, the in planta transformation did not employ antibiotic selection at an early stage, potentially resulting in chimerism (the presence of both transformed and non-transformed tissues within a single plant) among the obtained candidates [33]. This may have led to gene activation only in certain tissues, resulting in suboptimal production of dendrobine as well.

## 4. Materials and Methods

### 4.1. Plant Materials

In this study, *D. catenatum* and *D. nobile* were cultured on bark medium in a greenhouse at the Orchid Conservation and Research Center of Shenzhen. The greenhouse environment maintains a diurnal temperature range of 25–28 °C, a consistent relative humidity of 60%, and natural light. The natural light intensity in Shenzhen fluctuates throughout the day and across the year. The maximum illuminance can reach 10,000 lux in summer and 3500 lux in winter. The transgenic plants were grown in culture room conditions with a light intensity ranged from 12,880 to 12,980 lux, 25 °C, and 16 h light/8 h dark photo period.

### 4.2. Genomic PCR

For polymerase chain reaction (PCR) analysis of the transformed plants, genomic DNA (gDNA) was extracted. Approximately 100.0 mg of young leaves were collected and pulverized using a grinder after the addition of liquid nitrogen. Subsequently, gDNA was isolated using the Genomic DNA Purification kit (DP350-03, Tiangen, Beijing, China). The genomic DNA was extracted following the manufacturer’s instructions. Briefly, 400 µL of extraction buffer FGA and 6 µL of RNase A (10 mg/mL) were added and vortexed before incubation at room temperature for 10 min. Then, 130 µL of LP2 was added and mixed thoroughly by vortexing for 1 min at room temperature. Samples were spun at 12,000 rpm for 5 min at room temperature, and the supernatant was transferred into new 1.5 mL tubes. We added 750 µL of buffer LP3 and mixed thoroughly for 15 s to precipitate the DNA. Then, we transferred all of the substances into column CB3 and spun at 12,000 rpm for 30 s to enrich the DNA. We added 600 µL of washing buffer PW into the CB3 column and spun at 12,000 rpm for 30 s to purify the DNA. Finally, we added 50 µL of distilled water and spun at 12,000 rpm at room temperature for 2 min to elute the DNA. PCR was employed to amplify a 400 bp fragment of the *hptII* gene, indicating target fragment integration in the genome. Additionally, a 200 bp fragment of the *nptII* gene was amplified to exclude the possibility of bacterial contamination. A 200 bp fragment of *DcActin 7* (LOC110111141) served as a control for equal loading. The primers used were listed in Appendix A. The 2×EasyTaq^®^ PCR SuperMix (AS111-12, TansGen, Beijing, China) was utilized for amplification. The resulting PCR products were separated by electrophoresis on a 1.5% (*w*/*v*) agarose gel.

### 4.3. LC/MS

The dendrobine standard (CAS, 2115-91-5) was obtained from the National Institutes for Food and Drug Control (Beijing, China). The quantification of dendrobine content was primarily performed using LC/MS, with the analysis outsourced to Wuhan Pronetsbio Co., Ltd. (Wuhan, China) (http://www.pronetsbio.com/, accessed on March 1 2022). In brief, the experimental samples were pulverized into a fine powder, and approximately 0.25 g of the sample was weighed and transferred to a centrifuge tube. Subsequently, 10 mL of methanol solution containing 0.05% formic acid was utilized for extraction, and the total weight was recorded. The mixture underwent ultrasonic extraction at room temperature for 30 min. The weight was re-measured to account for any solvent loss due to evaporation, and the mixture was centrifuged at 5000 rpm to obtain the supernatant. The residue underwent two additional extractions with 10 mL and 5 mL of methanol solution containing 0.05% formic acid, respectively, after which all extracts were combined. Finally, the extract was diluted to an appropriate concentration prior to LC/MS analysis [34].

### 4.4. Dual-Luciferase Assay

The *MCT* promoter was cloned and inserted into the TQ379 vector [35], regulating the expression of firefly luciferase (LUC). The 35S-driven renilla luciferase (REN) served as an internal control. Protoplasts were extracted from young leaves of *D. catenatum* [36]. The protoplasts were transferred to a 2 mL EP tube to maintain a cell density of 2 × 10^6^ cells/mL in W5 medium (5 mM of glucose, 2 mM of MES, 154 mM of NaCl, 125 mM of CaCl_2_, 5 mM of KCl). A combination of 10 μg of the TQ379-proMCT reporter plasmid and the CRISPR/Act3.0-MCT-sgRNA effector plasmid was introduced into 100 μL of the protoplast suspension and gently mixed. As a control, a combination of 10 μg of the TQ379-proMCT reporter plasmid and the CRISPR/Act3.0 effector plasmid without the sgRNA was introduced into 100 μL of the protoplast suspension. Subsequently, an equal volume of PEG6000 solution was added, and the mixture was incubated at 25 °C for 30 min. The medium in the EP tube was removed, and 1 mL of freshly prepared W5 medium was introduced. The cells were further incubated at 25 °C in the dark for 12–18 h. Following incubation, the cells were centrifuged at low speed at room temperature (150 g, 2 min, brack 3 and acc 3), and the medium was removed. The Dual-Luciferase^®^ Reporter Assay System kit (Promega, USA, TM040) was utilized to measure the dual-luciferase activity.

### 4.5. Vector Construction

The SG-multigene overexpression vectors were constructed as described in our previously published article [17]. The sgRNAs (Appendix A) and the sgR2.0 backbone were synthesized by the Beijing Genomics Institute (BGI), sub-cloned using Phusion High-Fidelity DNA polymerase (New England Biolabs, F532L), and subsequently inserted into the pYPQ146-ZmUbi-tRNA vector (Addgene plasmid, 158404). Finally, the target DNA fragment was ligated into a plant expression vector containing the CRISPR/Act3.0 transcription activation domain (Addgene plasmid, 158408) via Gateway LR reaction. The resulting expression vector was electroporated into *Agrobacterium* for plant transformation.

### 4.6. Genetic Transformation

For transient expression, punctures were made on the abaxial side of young leaves using a syringe needle. The suspension was then slowly infiltrated into the leaves through these punctures using a needleless syringe. A minimum of 100 leaves were injected per group, with *Agrobacterium* strains carrying the empty vector serving as the control. Post-injection, the plants were cultivated under standard light conditions for subsequent sampling. The infiltrated leaves were collected at specific intervals: 6 h and 24 h for RNA extraction, and 5 days for dendrobine content analysis. The leaves underwent fixation at 100 °C, followed by drying at 80 °C for 3 days. Subsequently, the dendrobine content was quantified using LC/MS (Wuhan Pronetsbio Co., Ltd.). For in planta transformation [18], the resultant pellets were resuspended in MS medium containing 5.0% sucrose. The liquid medium for injection was prepared by adding 0.1% Silwet L-77 and 50.0 mg/L acetosyringone. For the infiltration experiment, 9-month-old *D. catenatum* seedlings were utilized. Initially, all visible buds and leaves were removed from the plants. *Agrobacterium* suspensions were then inoculated into the cut sites and stem nodes using a syringe, ensuring a thorough and uniform distribution. A minimum of five plants per group were injected, with *Agrobacterium* strains carrying the empty vector serving as the control. Post-inoculation, the plants were placed in darkness for one hour to enhance infection efficiency. The inoculated seedlings were then planted in a sterilized bark medium, with the surface covered by cling film to maintain high humidity for two days. Finally, the cling film was removed, and the seedlings were returned to normal cultivation conditions. New shoot regeneration was typically observed at the plant base and nodes approximately two weeks later.

### 4.7. Quantitative Real-Time PCR (qRT-PCR) Validation of Gene Expression

In this investigation, total RNA was extracted from frozen leaf samples using the Quick RNA Isolation Kit (Huayueyang, Beijing, China, 0416-50gk) to analyze the expression of target genes. Following extraction, 1.0 µg of total RNA was heat-treated to disrupt secondary structure, after which PrimeScript™ RT reagent Kit with gDNA Eraser (Takara, Dalian, China, RR047B) was added to perform the reverse transcription reaction. The resulting cDNA was appropriately diluted as a template and combined with Green qPCR MasterMix (Biomed, Beijing, China, MT521-03) and gene-specific primers. The primers utilized are listed in Appendix A. *D. catenatum Actin 7* (LOC110111141) served as an internal reference gene, with each sample prepared in triplicate. qRT-PCR was conducted using the ABI PRISM 7500 Fluorescent Quantitative PCR System (Thermo Fisher Scientific, Singapore), and the data obtained were analyzed to determine the relative gene expression levels using the comparative Ct method.

### 4.8. Data Analysis

Data were collected for each construct, and mean and standard deviation (SD) values were calculated. The data were analyzed, and figures were generated using the GraphPad Prism 8 software package (La Jolla, CA, USA). The statistical significance of differences (*p* < 0.05) compared to control samples was expressed using an unpaired Student’s *t*-test, with a minimum of three replicates per construct.

## Figures and Tables

**Figure 1 ijms-26-01487-f001:**
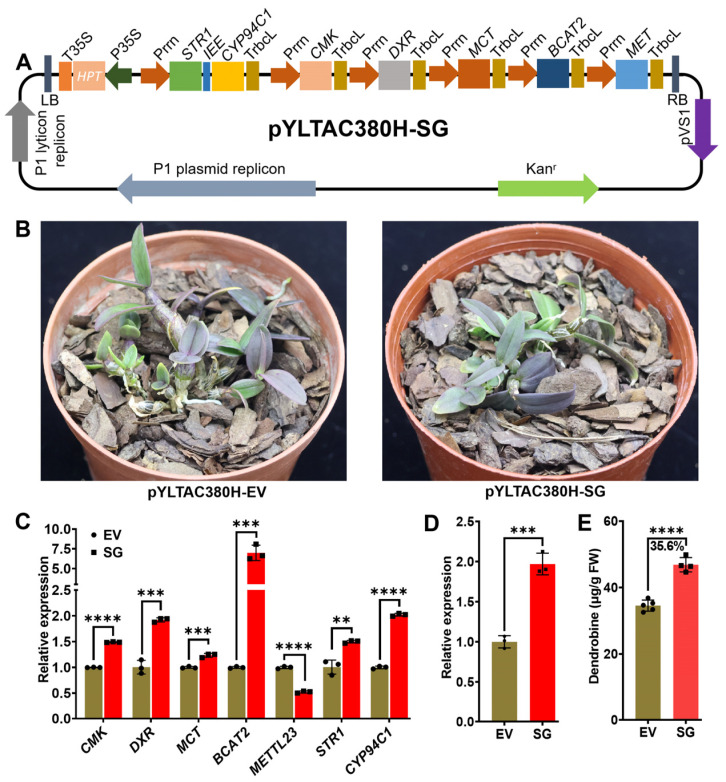
Overexpression of multiple stacked genes enhanced dendrobine production in *D. catenatum* transgenic plants. (**A**) Physical map of the multigene constructs utilized for integration and expression of the synthetic operons (adapted from [17]). (**B**) Transgenic *D. catenatum* plantlets with SG-multigene and empty vector control were grown in culture room conditions with a light intensity ranging from 12,880 to 12,980 lux, 25 °C, and with an 16 h light/8 h dark photo period. (**C**) Expression analysis of individual genes in SG-multigene transgenic *D. catenatum* by qRT-PCR. EV transgenic plants served as the control. (**D**) Expression analysis of the *FPPS* marker gene in transgenic plants. *FPPS* expression in EV transgenic plants were served as the control. (**E**) Quantification of dendrobine content in the leaves of transgenic *D. catenatum* by LC/MS. Asterisks indicate statistical significance based on Student’s *t*-test. ** *p* ≤ 0.01; *** *p* ≤ 0.001; **** *p* ≤ 0.0001.

**Figure 2 ijms-26-01487-f002:**
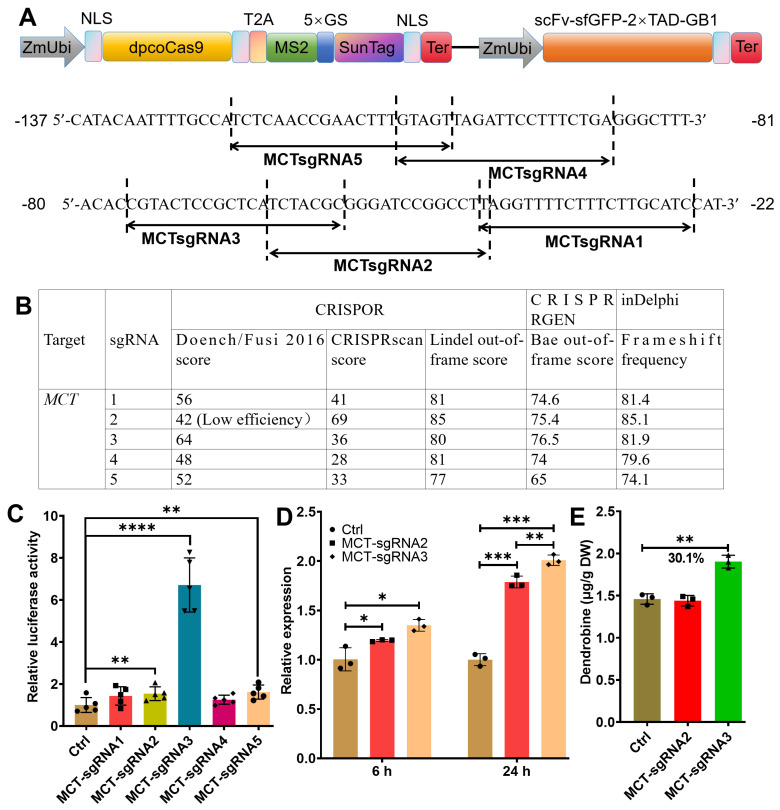
Highly efficient sgRNA prediction method. (**A**) Schematic representation of sgRNA locations in the *MCT* promoter. Numbers indicate upstream positions starting from initiation codon ATG. (**B**) Efficiency rating scale for sgRNAs by CRISPOR. (**C**) sgRNA-guided transcriptional activation of *MCT* in protoplasts 12 h post transformation. Vector without sgRNA co-transformed with REN/LUC reporter served as the control (Ctrl). (**D**) sgRNA-guided transcriptional activation of *MCT* in transiently infiltrated leaves of 17-month-old *D. catenatum*. Data were measured at 6 and 24 h post transformation. Transiently transformed leaves with empty vector served as the control. The plants were grown in green house conditions with diurnal temperature range of 25–28 °C, a consistent relative humidity of 60%, and natural light. (**E**) LC/MS quantification of dendrobine content in *D. catenatum* leaves five days after infiltration. Asterisks indicate statistical significance based on Student’s *t*-test. * *p* ≤ 0.05; ** *p* ≤ 0.01; *** *p* ≤ 0.001, **** *p* ≤ 0.0001.

**Figure 3 ijms-26-01487-f003:**
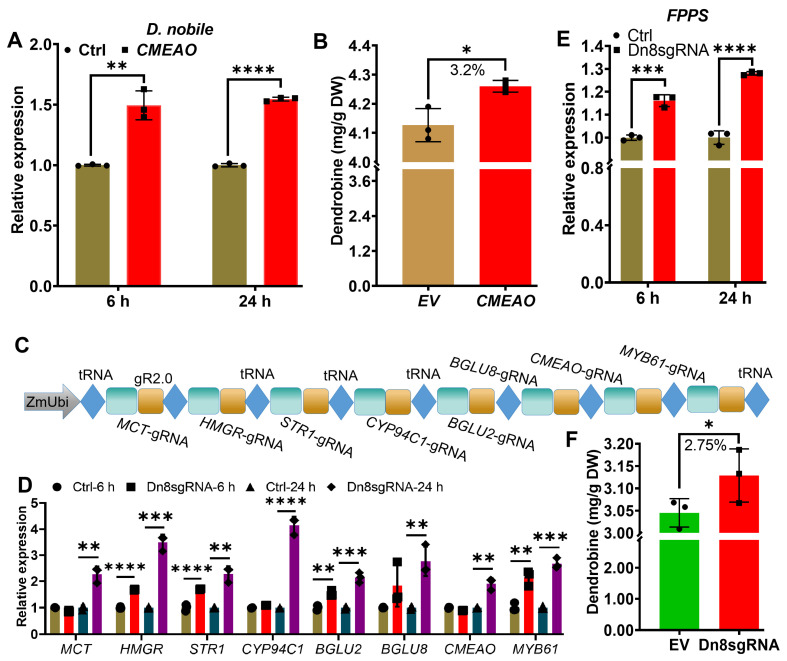
Enhancement of dendrobine content in *D. nobile* by CRISPR/Act3.0. (**A**) Gene expression of *CMEAO* in leaves of *D. nobile* (17-month-old) grown in green house conditions. Leaves infiltrated with Agrobacterium carrying vector lack of sgRNA served as the control (Ctrl). (**B**) Quantification of dendrobine content in *D. nobile* leaves five days after infiltration by LC/MS. (**C**) Schematic representation of the sgRNA array targeting eight endogenous genes. (**D**) Validation of gene activation by CRlSPR/Act3.0 in transiently infiltrated leaves of *D. nobile* (18-month-old). Leaves infiltrated with Agrobacterium carrying vector lack of sgRNA served as the control (Ctrl). (**E**) *FPPS* expression analysis to demonstrate activation of dendrobine synthesis pathway. Ctrl: Samples collected from EV transformed leaves 6 and 24 h after infiltration. (**F**) Quantification of dendrobine content by LC/MS in *D. nobile* expressing the activation unit five days after infiltration. Asterisks indicate statistical significance based on Student’s *t*-test. * *p* ≤ 0.05; ** *p* ≤ 0.01; *** *p* ≤ 0.001; **** *p* ≤ 0.0001.

**Figure 4 ijms-26-01487-f004:**
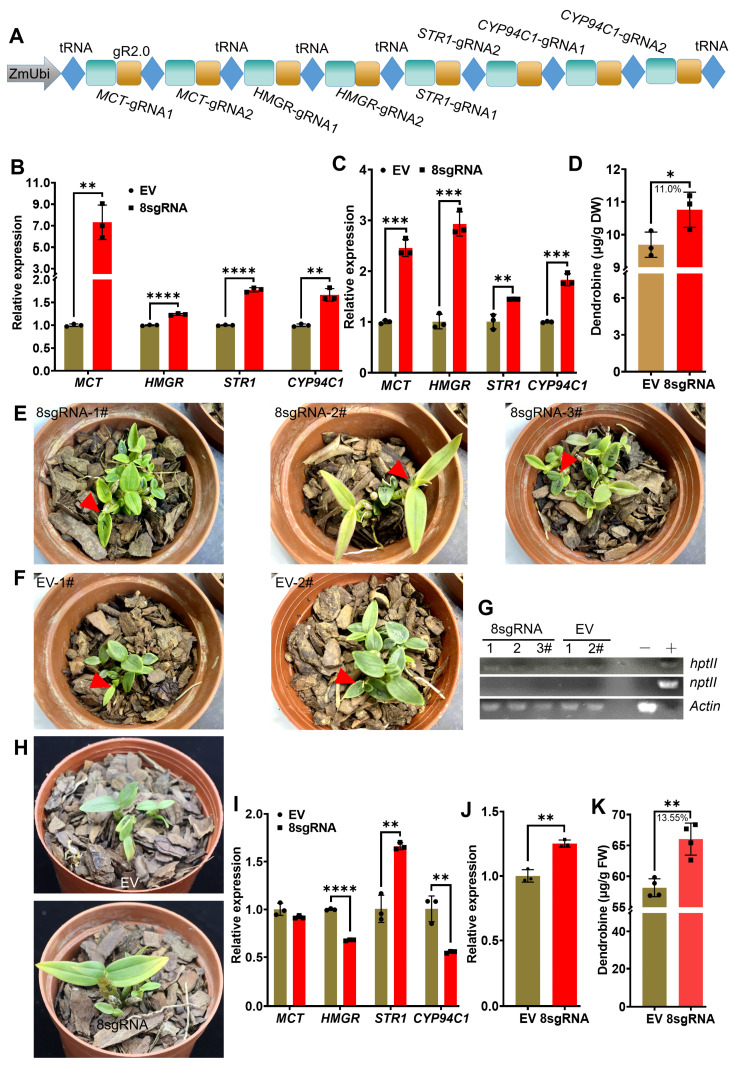
Enhancement of dendrobine content in *D. catenatum* by CRISPR/Act3.0. (**A**) Schematic representation of the combination of the four endogenous targets with two sgRNAs targeting each gene, respectively. (**B**) Verification of gene activations by CRISPR/Act3.0 in protoplasts 12 h post transformation. (**C**) Gene activations by CRlSPR/Act3.0 in leaves of *D. catenatum* (26-month-old) 24 h post infiltration. The plants were grown in green house conditions with diurnal temperature range of 25–28 °C, a consistent relative humidity of 60%, and natural light. (**D**) Quantification of dendrobine content in the leaves of *D. catenatum* expressing the CRISPR/Act3.0-8sgRNA using LC/MS five days post infiltration. (**E**,**F**) Phenotypic characteristics of CRISPR/Act3.0-8sgRNA stably expressing transgenic *D. catenatum* plantlets (13-month-old). The transgenic plants were grown in culture room conditions with a light intensity ranging from 12880 to 12980 lux, 25 °C, and with a 16 h light/8 h dark photo period. The positive transgenic plants were labeled with a red triangle. (**G**) Molecular characterization of the *hptII* transgene in *D. catenatum* transgenic plants. “−” indicates wild-type plants while “+” represents the empty vector plasmid used as a positive control. (**H**) Collection of transgenic *D. catenatum* plants and subsequent replanting in bark pots (16-month-old). (**I**) Expression analysis of individual genes in transgenic plants. (**J**) Expression of the *FPPS* marker gene in transgenic plants. (**K**) Quantification of dendrobine content in transgenic plants using LC/MS. Asterisks indicate statistical significance based on Student’s *t*-test. * *p* ≤ 0.05; ** *p* ≤ 0.01; *** *p* ≤ 0.001; **** *p* ≤ 0.0001.

## Data Availability

The authors confirm that all data from this study are available and can be found in this article and in Appendix A.

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
