# Peer review of "Enhancement of Dendrobine Production by CRISPR/Act3.0-Mediated Transcriptional Activation of Multiple Endogenous Genes in Dendrobium Plants"

_ijms, 2025, doi:10.3390/ijms26041487_

Round 1
Reviewer 1 Report
Comments and Suggestions for Authors
The study entitled “Enhancement of Dendrobine Production by CRISPR/Act3.0-Mediated Transcriptional Activation of Multiple Endogenous Genes in Dendrobium Plants” explored two methods—ectopic overexpression of genes and activation of endogenous genes using CRISPR/Act3.0—to increase dendrobine, a valuable medicinal compound, in Dendrobium plants. This work is promising because it is the first time the CRISPR/Act3.0 approach has been employed in Dendrobium species. However, several mistakes are found in the experimental design and the claims can not be supported by the data.
Major concerns:
In general, there is a lot of variation in the dendrobine content D. catenatum for each experiment in the data shown: Figure 1E, 2E, 3H, 4D and 4K, (1.5 ug/g Dw to more than 3000 ug/gDw). Is this an effect of the age of the plant? To offer consistency to the manuscript, the experiments should be carried out in plants of the same age (or at least as much similar as possible). Changes in the age of the plant produce changes in the metabolic processes so is difficult to compare the results between experiments.
Section 2.2:
Figure 2A: What is the reference point in (+1) in the promoter? Is it from the TSS or the ATG? It’s a common good practice in the field of CRISPRa to first determine which is the optimal distance from the TSS. The manuscript should include a few lines to specify how the gRNAs were designed.
Figure 2C: Statistical analysis is missing.
Line 121 to line 125:More information is needed about the experimental set up. Were sgRNA transiently expressed or stable transgenic lines were generated? Besides, gRNA3 was much more efficient in gene activation (Fig2C). However, in Fig2D, there is not an obvious difference in gene activation between sgRNA2 and sgRNA3. Given that sgRNA2 is not considered as an efficient sgRNA, there is an inconsistency within the results.
Figure 2D: Here, there is no specification about the control employed to compare the sample. Since is a time course experiment, two different controls are necessary: one at 6 h and another at 24 h. This figure contradicts the conclusion of 2E, since the relative expression of the gene MCT at 24h employing sgRNA2 and sgRNA3, although statistically different, is not biologically relevant. The transcriptional gene activation is about 1.8 fold with sgRNA2 and 2 fold with sgRNA3 and this minimal difference does not correlate with the metabolic results of Figure 2E.
Figure 2E: The scale of the Y axis seems to be manipulated to increase the magnitude of the increment dendrobine. Note that the y axis is not set on 0 (as it is the case for other figures), but starts at 1.2. The increment goes from around 1.4 to 1.9 ug/g Dw, which is notable but It will be more fair to the reader if the figures include the % of increment.
Section 2.3
Line 143 to 148: More information is needed about the experimental set up. Were genes transiently expressed or stable transgenic lines were generated? Were the controls also infiltrated? Also, usually, mechanical wounding (such as puncturing the leaves with needles in infiltration) in triggers the Jasmonic Acid signalling. JA is a master regulator of specialized metabolism. pathways, including alkaloid production (https://doi.org/10.3389/fpls.2022.941231). Thus, proper control are needed to exclude that dendrobine synthesis is not triggered by wounding itself. In the current state of the manuscript, it is unclear if the effect of the tissue damage in the experiment was taken into account.
Figure 3A and C: No statistical analysis was employed here. Additionally, the way to show the error bar is not consistent with the other graphs and the points for each replicate are missing. Without proper data analysis, the claim in lines 142-144 cant be supported, as it not evident that CMEAO was overexpressed. According to the legend, the control employed to compare the gene expression was always at 6h. If the experiment compares different time points, controls for 12 and 24 hours should be included.
Figure 3B: Same comment as Figure 2E.
Figure 3D: lack of consistency in the graph representation: No completed error bars and no individual points. It was claimed that LC/MS is a better approach to measure dendrobine than HPLC. Why the content it was measured with the HPLC just for this experiment?
Figure 3F: The control for 24 hours is not included, so the 24h point can not be included in the analysis of the data.
Figure 3G: Same as 3F. If no sgRNAs for FPPS were included in the construct, how the untargeted activation of FPPS can be explained? Can the pathway be activated by an alternative route, such us mechanical wounding? This observation has to be further discussed.
Figure 3H: Same comment as Figures 2E and 3B.
Section 2.4:
It was never mentioned which gRNAs were employed in the experiments represented in the Figure 4.
Figure 4A only shows a single gRNA x gene.
Figure 4B: No statistical analysis was employed here. Additionally, the way to show the error bar is not consistent with the other graphs and the points for each replicate are missing. Without proper data analysis, the claim in lines 187-188 cant be supported, as it not evident that some genes were overexpressed.
Figure 4C: Lack of consistency in the graph representation (error bars and missing individual points).
Figure 4D: Same comment as Figures 2E, 3B and 3H.
Line 200 to 203: The manuscript mentions a potential silencing, but also a slight upregulation of FPPS. Why is this relevant? Please, explain further.
Figure 4I: Which gene was employed as a control? Only one control appears but four different comparisons are performed. No statistics were performed.
Figure 4K: Same comment as Figures 2E, 3B, 3H and 4D.
The discussion of this manuscript must be re-evaluated. For example, given that there is a discrepancy for the activation efficiency of sgRNA3 in protoplast vs leaves, the discussion should offer possible explanations for this observation. Furthermore, the work with D. catenatum is not discussed. Which conclusions and limitations were observed with this species? Finally, in line 270-273 the lack of use of antibiotics is discussed. is this a normal procedure in the dendrobium transformation?
Minor
In the introduction, the state of art of the CRISPRa field is not well covered. For example, previous work of metabolic engineering in other species (see 10.1016/j.copbio.2022.102856 and references there).
The section 2.3 was fully performed in D. nobile, but the test of BGLU genes was performed in D. catenatum. Consider moving these results to another section.
Finally, the Figure captions should be more complete, including the details of the controls. Also, the statistical test used was not mentioned with except for Figure 1
Author Response
The study entitled “Enhancement of Dendrobine Production by CRISPR/Act3.0-Mediated Transcriptional Activation of Multiple Endogenous Genes in Dendrobium Plants” explored two methods—ectopic overexpression of genes and activation of endogenous genes using CRISPR/Act3.0—to increase dendrobine, a valuable medicinal compound, in Dendrobium plants. This work is promising because it is the first time the CRISPR/Act3.0 approach has been employed in Dendrobium species. However, several mistakes are found in the experimental design and the claims can not be supported by the data.
A: Thank you for your comments. Your input is greatly appreciated. We clarified the data according to your suggestions.
Major concerns:
In general, there is a lot of variation in the dendrobine content in D. catenatum for each experiment in the data shown: Figure 1E, 2E, 3H, 4D and 4K, (1.5 μg/g DW to more than 3000 μg/g DW). Is this an effect of the age of the plant? To offer consistency to the manuscript, the experiments should be carried out in plants of the same age (or at least as much similar as possible). Changes in the age of the plant produce changes in the metabolic processes so is difficult to compare the results between experiments.
A: Thank you for your great question. We described in Line288~300: The dendrobine content in the 1-year-old greenhouse-grown D. catenatum plants used for transient infiltration of leaves ranged from 1.5 µg/g DW (Figure 2E) to 11.5 µg/g DW (Figure 4D), as determined by more precise LC/MS analysis. However, the relative quantification of dendrobine by HPLC reaches approximately 3000 µg/g DW. This study employs HPLC to compare the relative quantification of dendrobine and to screen candidate genes, while LC/MS is used for the precise determination of dendrobine content. Moreover, D. catenatum plants grown in the culture room accumulated higher levels of dendrobine compared to those grown in the greenhouse, ranging from 35 µg/g FW (Figure 1E) to 70 µg/g FW (Figure 4K), possibly due to the varied conditions. The greenhouse environment maintains a diurnal temperature range of 25-28°C, a consistent relative humidity of 60%, and natural light. The natural light intensity fluctuates throughout the day and across the year. The maximum illuminance can reach 10,000 lux in summer and 3,500 lux in winter. The transgenic plants were grown in culture room conditions with a light intensity ranged from 12,880-12,980 lux, 25°C, and 16-h light/8-h dark photo period. Nevertheless, the overall enhancement tendency of CRISPR-Act3.0 on endogenous gene expression and thus boosting dendrobine production remains consistent.
Section 2.2:
Figure 2A: What is the reference point in (+1) in the promoter? Is it from the TSS or the ATG? It’s a common good practice in the field of CRISPRa to first determine which is the optimal distance from the TSS. The manuscript should include a few lines to specify how the gRNAs were designed.
A: Thank you for your great suggestion. We give the reference point in (+1) in the promoter in Figure 2A in Line121~122: Because in many cases the transcription starting site (TSS) is hard to precisely predict, the candidate sgRNAs were designed within 200 base pairs upstream of the initiation codon ATG. We described the sgRNA design methods in Line 318~323: CRISPRa efficiency is sensitive to the sgRNA targeting site relative to the transcription start site (TSS), and effective sgRNAs are often located within a 200 bp region upstream of the TSS. However, precise annotation of the TSS in the promoter is still challenging due to the complex architecture of promoter sequences. For this reason, we design sgRNAs for CRISPR-Act3.0 targeting the proximal promoter between -200 bp and the initiation codon ATG, then evaluate using CRISPOR, verify using protoplasts, or design multiple sgRNAs for one specific gene.
Figure 2C: Statistical analysis is missing.
A: Thank you for your suggestion. We added statistical analysis for Figure 2C.
Line 121 to line 125: More information is needed about the experimental set up. Were sgRNA transiently expressed or stable transgenic lines were generated? Besides, gRNA3 was much more efficient in gene activation (Fig2C). However, in Fig2D, there is not an obvious difference in gene activation between sgRNA2 and sgRNA3. Given that sgRNA2 is not considered as an efficient sgRNA, there is an inconsistency within the results.
A: Thank you for your comments and suggestions. We described in Line127~128: Subsequently, we selected MCT-sgRNA3 and MCT-sgRNA2 for further testing in transiently infiltrated Dendrobium leaves. For the consistency, we described in Line307~315: According to the protocol, protoplasts were cultured for 12 hours before gene expression analysis. The activation of sgRNA2, sgRNA3, and sgRNA5 was detected, with sgRNA3 exhibiting the highest activity. Subsequently, sgRNA3 was further tested in transiently infiltrated leaves, with sgRNA2 serving as the control. The activation of MCT expression was analyzed at 6 and 24 hours, while dendrobine content was measured five days after infiltration. The activation of MCT expression was assessed at different time points using both the protoplast and transient leaf expression systems. It is evident that the final concentration of dendrobine results from the cumulative expression of MCT during the five-day culture, and sgRNA2 was consistently less effective than sgRNA3 across all the time points. Therefore, sgRNAs with the highest scores are recommended.
Figure 2D: Here, there is no specification about the control employed to compare the sample. Since it is a time course experiment, two different controls are necessary: one at 6 h and another at 24 h. This figure contradicts the conclusion of 2E, since the relative expression of the gene MCT at 24h employing sgRNA2 and sgRNA3, although statistically different, is not biologically relevant. The transcriptional gene activation is about 1.8 fold with sgRNA2 and 2 fold with sgRNA3 and this minimal difference does not correlate with the metabolic results of Figure 2E.
A: Thank you for your excellent comments. In Figure 2D, it is indeed a time course experiment. Two different controls are necessary: one at 6 h and another at 24 h. We showed the controls for 6 h and 24 h, respectively, in this revised version. Because the Ctrls were standardized to “1”, we used one Ctrl in the last version of the manuscript. Yes, we agree with the reviewer that it is better to give all of the Ctrls and we do so for the other figures as well, including Figure 2D, Figure 3A, Figure 3D, Figure 3E, and Figure S2A. For your concerns over the conclusion of Figure 2E, we described in Line310~314: The activation of MCT expression was analyzed at 6 and 24 hours, while dendrobine content was measured five days after infiltration. The activation of MCT expression was assessed at different time points using the transient leaf expression system. It is evident that the final concentration of dendrobine results from the cumulative expression of MCT during the five-day culture, and sgRNA2 was consistently less effective than sgRNA3.
Figure 2E: The scale of the Y axis seems to be manipulated to increase the magnitude of the increment dendrobine. Note that the y axis is not set on 0 (as it is the case for other figures), but starts at 1.2. The increment goes from around 1.4 to 1.9 µg/g DW, which is notable but It will be more fair to the reader if the figures include the % of increment.
A: Thank you for your suggestion. We re-scaled the Y axis starting from 0, and showed the 30.1% increment in Figure 2E.
Section 2.3
Line 143 to 148: More information is needed about the experimental set up. Were genes transiently expressed or stable transgenic lines were generated? Were the controls also infiltrated? Also, usually, mechanical wounding (such as puncturing the leaves with needles in infiltration) in triggers the Jasmonic Acid signalling. JA is a master regulator of specialized metabolism. pathways, including alkaloid production (https://doi.org/10.3389/fpls.2022.941231). Thus, proper control are needed to exclude that dendrobine synthesis is not triggered by wounding itself. In the current state of the manuscript, it is unclear if the effect of the tissue damage in the experiment was taken into account.
A: Thank you for your great suggestion. Yes, we have taken the mechanical damage possibly originates from the infiltration process into consideration, and infiltrated the empty vector control into the leaves as well. We clarified this in Line165~169: To investigate this, we transiently overexpressed CMEAO (Copper Methylamine Oxidase), a gene previously identified as a positive regulator for dendrobine synthesis, in leaves of D. nobile (Figure 3A). As anticipated, the overexpression of CMEAO elevated dendrobine level in D. nobile in comparison to empty vector infiltrated control, suggesting that the function of the CMEAO gene is conserved across Dendrobium species (Figure 3B).
Figure 3A and C: No statistical analysis was employed here. Additionally, the way to show the error bar is not consistent with the other graphs and the points for each replicate are missing. Without proper data analysis, the claim in lines 142-144 cant be supported, as it not evident that CMEAO was overexpressed. According to the legend, the control employed to compare the gene expression was always at 6h. If the experiment compares different time points, controls for 12 and 24 hours should be included.
A: Thank you for your suggestion. We performed statistical analysis in Figure 3A and Figure S2A (Originally Figure 3C). We showed the complete error bar and points for each replicate. The statistical analysis demonstrated the significant upregulation of CMEAO in all the time points tested. Because the Ctrls were standardized to “1”, we used one Ctrl in the last version of the manuscript. Yes, we agree with the reviewer that it is better to give all of the Ctrls and we do so for the other figures as well, including Figure 2D, Figure 3A, Figure 3D, Figure 3E, and Figure S2A.
Figure 3B: Same comment as Figure 2E.
A: Thank you for your suggestion. We clarified the data.
Figure 3D: lack of consistency in the graph representation: No completed error bars and no individual points. It was claimed that LC/MS is a better approach to measure dendrobine than HPLC. Why the content it was measured with the HPLC just for this experiment?
A: Thank you for your suggestion. We clarified the data. We moved the original Figure 3C and D into Figure S2A and B. This study employs HPLC, which is more cost effective, to compare the relative quantification of dendrobine and to screen out candidate genes, while LC/MS is used for the precise determination of dendrobine content.
Figure 3F: The control for 24 hours is not included, so the 24h point can not be included in the analysis of the data.
A: Thank you for your suggestion. We clarified the data by providing all of the controls including the control for 24 hours.
Figure 3G: Same as 3F. If no sgRNAs for FPPS were included in the construct, how the untargeted activation of FPPS can be explained? Can the pathway be activated by an alternative route, such us mechanical wounding? This observation has to be further discussed.
A: Thank you for your great suggestion. We clarified the data by providing all of the controls including the control for 24 hours. For FPPS activation, we explained in Line335~340: Even though the whole transcription profile has not been checked, the increased production of dendrobine is thought to be resulted from the upregulation of multiple positive genes in the synthesis pathway. FPPS encodes a key enzyme responsible for the precursor synthesis of dendrobine. Overexpression of FPPS is corresponding to enhanced dendrobine production [30]. Therefore, the upregulation of FPPS indicates the activation of the synthesis pathway.
Figure 3H: Same comment as Figures 2E and 3B.
A: Thank you for your suggestion. We clarified the data.
Section 2.4:
It was never mentioned which gRNAs were employed in the experiments represented in the Figure 4. Figure 4A only shows a single gRNA per gene.
A: Thank you for your suggestion. We added two sgRNA for each gene in Figure 4A, and provided sgRNA sequences in Table S2.
Figure 4B: No statistical analysis was employed here. Additionally, the way to show the error bar is not consistent with the other graphs and the points for each replicate are missing. Without proper data analysis, the claim in lines 187-188 cant be supported, as it not evident that some genes were overexpressed.
A: Thank you for your great suggestion. We performed statistical analysis in Figure 4B. We visualized the complete error bars and points for each replicate. The statistical analysis demonstrated the significant upregulation of each of the genes tested.
Figure 4C: Lack of consistency in the graph representation (error bars and missing individual points).
A: Thank you for your suggestion. We performed statistical analysis in Figure 4C and visualized the complete error bars and points for each replicate.
Figure 4D: Same comment as Figures 2E, 3B and 3H.
A: Thank you for your suggestion. We clarified the data.
Line 200 to 203: The manuscript mentions a potential silencing, but also a slight upregulation of FPPS. Why is this relevant? Please, explain further.
A: Thank you for your question. We clarified this issue in Line226~230: While no visual differences were apparent, the analysis of the targeted gene expression revealed a significant increase in STR1, whereas the other three genes did not exhibit similar upregulation (Figure 4I), suggesting potential gene silencing. The downregulation of the other two genes compromises the contribution of STR1 to dendrobine synthesis. Consistent with this finding, a slight upregulation of the FPPS gene was also observed (Figure 4J), which was accompanied by suboptimal improvements in dendrobine production (Figure 4K).
Figure 4I: Which gene was employed as a control? Only one control appears but four different comparisons are performed. No statistics were performed.
A: Thank you for your good suggestion. We clarified the data.
Figure 4K: Same comment as Figures 2E, 3B, 3H and 4D.
A: Thank you for your good suggestion. We clarified the data.
The discussion of this manuscript must be re-evaluated. For example, given that there is a discrepancy for the activation efficiency of sgRNA3 in protoplast vs leaves, the discussion should offer possible explanations for this observation. Furthermore, the work with D. catenatum is not discussed. Which conclusions and limitations were observed with this species? Finally, in line 270-273 the lack of use of antibiotics is discussed. is this a normal procedure in the dendrobium transformation?
A: Thank you for your great suggestion. For your concern over the protoplast vs leaves, we explained in Line308~316: According to the protocol, protoplasts were cultured for 12 hours before gene expression analysis. The activation of sgRNA2, sgRNA3, and sgRNA5 was detected, with sgRNA3 exhibiting the highest activity. Subsequently, sgRNA3 was further tested in transiently infiltrated leaves, with sgRNA2 serving as the control. The activation of MCT expression was analyzed at 6 and 24 hours, while dendrobine content was measured five days after infiltration. The activation of MCT expression was assessed at different time points using both the protoplast and transient leaf expression systems. It is evident that the final concentration of dendrobine results from the cumulative expression of MCT during the five-day culture, and sgRNA2 was consistently less effective than sgRNA3. Therefore, sgRNAs with the highest scores are recommended. Furthermore, the work with D. catenatum is discussed in Line264~276: The study applied CRISPR/Act3.0 for the first time to probe the potential of this new technology in Dendrobium species. Endogenous activation of dendrobine synthesis genes by CRISPR/Act3.0 resulted in enhanced dendrobine production, with more profound effects observed in D. catenatum. Through expression manipulation of multiple genes in the synthesis pathway, an increase of up to 35.6% in dendrobine production was achieved. This will contribute to increasing the added value of D. catenatum and lay a solid foundation for further improvements. The major challenge, however, is still the tedious work of genetic transformation and the long duration of obtaining transformants. So far, the most common method for genetic transformation of Dendrobium species is mediated by Agrobacterium infiltration of protocorm. Yet, no standard guidelines are available. More than ten rounds of antibiotic selection are needed to obtain transformants. To improve transformation efficiency and accelerate the process, in planta transformation was developed previously and utilized in this study to create stable transgenic D. catenatum plants. Because no antibiotic selection was applied in the early stages of transformation, chimerism might occur. Further evaluation of the chimerism and the physiological changes in the transgenic plants is needed.
Minor
In the introduction, the state of art of the CRISPRa field is not well covered. For example, previous work of metabolic engineering in other species (see 10.1016/j.copbio.2022.102856 and references there).
A: Thank you for your great suggestion. We added the advancements in metabolic manipulation by CRISPRa in Line59~60 and cited the paper there.
The section 2.3 was fully performed in D. nobile, but the test of BGLU genes was performed in D. catenatum. Consider moving these results to another section.
A: Thank you for your great suggestion. We moved this part into FigureS2.
Finally, the Figure captions should be more complete, including the details of the controls. Also, the statistical test used was not mentioned with except for Figure 1
A: Thank you for your good suggestions. We amended the Figure captions including the details of the controls and provided statistical test used in all the figures.

Reviewer 2 Report
Comments and Suggestions for Authors
The manuscript addressed Boosting Dendrobine Production through CRISPR/Act3.0-Mediated Transcriptional Activation of Multiple Endogenous Genes in Dendrobium Plants. The manuscript is generally good However, the following points need to be addressed by the authors before acceptance.
1- The abstract should contain specific key data.
2- The significance of the study should be given at the end of the abstract.
3- The novelty of CRISPR/Act3.0 approach could be emphasized more strongly.
4-The introduction does not clearly articulate the specific research gap the study aims to address. Elaborate more on the motivation of the study.
5-Results: The section abruptly starts with "Previously, we stacked multiple genes..." without clearly connecting to specific findings.
6-Discussion: Elaborate more on METTL23 downregulation, any potential impact of RNA interference or positional effects?
7- How does variability in dendrobine content impact the overall conclusion of the study ?
8- Elaborate more on how chimerism might affect gene expression and alkaloid biosynthesis with relevant citation.
9-Materials and Methos: Provide more details of the greenhouse conditions, such as the specific range of natural light intensity.
10- Include more details about the genomic DNA extraction process, such as specific incubation times and temperatures during each step.
11- Provide details about the controls used in dual luciferase assay.
12- The entire manuscript should be revised by a native English language expert.
Author Response
The manuscript addressed Boosting Dendrobine Production through CRISPR/Act3.0-Mediated Transcriptional Activation of Multiple Endogenous Genes in Dendrobium Plants. The manuscript is generally good. However, the following points need to be addressed by the authors before acceptance.
A: Thank you for your comments.
1.The abstract should contain specific key data.
A: Thank you for your good suggestion. We added the key data in the Abstract: These methods enhanced dendrobine production in transiently infiltrated leaves by 30.1% and transgenic plants by 35.6%.
2.The significance of the study should be given at the end of the abstract.
A: Thank you for your great suggestion. We highlighted the significance in the Abstract: The study is the first to apply CRISPR/Act3.0 to Dendrobium orchids, enhancing dendrobine production and thus laying a solid foundation for further improvements.
3.The novelty of CRISPR/Act3.0 approach could be emphasized more strongly.
A: Thank you for your suggestion. We emphasized the novelty of CRISPR/Act3.0 approach in the Abstract: CRISPR/Act3.0 is a recently developed technique that demonstrates high efficiency in model plant species, including rice, maize, and Arabidopsis. By combining CRISPR with transcriptional regulatory modules, activation of multiple endogenous genes in the metabolic pathway can be achieved. The successful application in orchid molecular breeding reveals promising potential for further exploration.
4. The introduction does not clearly articulate the specific research gap the study aims to address. Elaborate more on the motivation of the study.
A: Thank you for your good suggestion. We elaborated on the current research gap in Line52~55: The major source of dendrobine to date is still extraction from D. nobile, which is insufficient due to the long growth period and low concentration. Therefore, improving the production of dendrobine in Dendrobium plants and diversifying the source materials is of great significance for further development and utilization. We described our goal of the current study in Line69~76: Our previous study reconstituted seven dendrobine biosynthesis-related genes and generated transgenic plants in D. catenatum [17]. In this investigation, we further assessed gene expression and dendrobine content in these transgenic plants. Furthermore, we applied CRISPR/Act3.0 to achieve multiple endogenous gene activations in the dendrobine synthesis pathway of D. nobile and D. catenatum. Additionally, utilizing the in planta transformation method we previously developed for D. catenatum [18], we generated CRISPR/Act3.0 transgenic plants. This study applies newly developed technologies to perennial and transformation-recalcitrant Dendrobium orchids, establishing a foundation for molecular breeding in these species.
5. Results: The section abruptly starts with "Previously, we stacked multiple genes..." without clearly connecting to specific findings.
A: Thank you for your great suggestion. We provided more details about the specific finding in Line80~85: Currently, genetic manipulation of orchids is challenging due to their long growth periods and recalcitrance to regeneration. Although there are plenty of studies using orchid materials, research involving transgenic orchids is sporadic. To enhance dendrobine production in D. catenatum, we previously stacked multiple genes (SG, seven genes) involved in the dendrobine synthesis pathway into one vector (Figure 1A) and transformed it into D. catenatum. Several transgenic seedlings were obtained with activated Farnesyl Pyrophosphate Synthase (FPPS) expression, indicating enhanced dendrobine production [17].
6. Discussion: Elaborate more on METTL23downregulation, any potential impact of RNA interference or positional effects?
A: Thank you for your suggestion. We discussed the possible causes of the down regulation of METTL23 in Line278~282: In our SG-multigene transgenic Dendrobium plants, we observed a downregulation of METTL23. This downregulation may be attributed to the repeated use of Prrn promoters and Trbcl terminators in the constructs, as well as the potential impacts of RNA interference and positional effects. Further investigation is needed to clarify these factors.
7.How does variability in dendrobine content impact the overall conclusion of the study ?
A: Thank you for your great question. The dendrobine content in the 1-year-old greenhouse-grown D. catenatum plants used for transient infiltration of leaves ranged from 1.5 µg/g DW (Figure 2E) to 11.5 µg/g DW (Figure 4D), as determined by more precise LC/MS analysis. However, the relative quantification of dendrobine by HPLC reaches approximately 3000 µg/g DW. This study employs HPLC to compare the relative quantification of dendrobine and to screen candidate genes, while LC/MS is used for the precise determination of dendrobine content. Moreover, D. catenatum plants grown in the culture room accumulated higher levels of dendrobine compared to those grown in the greenhouse, ranging from 35 µg/g FW (Figure 1E) to 70 µg/g FW (Figure 4K). Nevertheless, the overall enhancement tendency of CRISPR-Act3.0 on endogenous gene expression and thus boosting dendrobine production remains consistent.
8.Elaborate more on how chimerism might affect gene expression and alkaloid biosynthesis with relevant citation.
A: Thank you for your great suggestion. We articulated this matter in Line347~351: However, the in planta transformation did not employ antibiotic selection at an early stage, potentially resulting in chimerism (the presence of both transformed and non-transformed tissues within a single plant) among the obtained candidates [33]. This may have led to gene activation only in certain tissues, resulting in suboptimal production of dendrobine as well.
9. Materials and Methos: Provide more details of the greenhouse conditions, such as the specific range of natural light intensity.
A: Thank you for your excellent suggestion. We detailed the environmental conditions in Line355~359: The greenhouse environment maintains a diurnal temperature range of 25-28°C, a consistent relative humidity of 60%, and natural light. The natural light intensity in Shenzhen fluctuates throughout the day and across the year. The maximum illuminance can reach 10,000 lux in summer and 3,500 lux in winter. The transgenic plants were grown in culture room conditions with a light intensity ranged from 12,880-12,980 lux, 25°C, and 16-h light/8-h dark photo period.
10.Include more details about the genomic DNA extraction process, such as specific incubation times and temperatures during each step.
A: Thank you for your great suggestion. We detailed the DNA extraction method in Line364~372: The genomic DNA was extracted following the manufacturer's instructions. Briefly, 400 µL of extraction buffer FGA and 6 µL of RNase A (10 mg/mL) were added and vortexed before incubation at room temperature for 10 minutes. Then, 130 µL of LP2 was added and mixed thoroughly by vortexing for 1 minute at room temperature. Samples were spun at 12,000 rpm for 5 minutes at room temperature, and the supernatant was transferred into new 1.5 mL tubes. Add 750 µL of buffer LP3 and mix thoroughly for 15 seconds to precipitate the DNA. Transfer all of the substances into column CB3 and spin at 12,000 rpm for 30 seconds to enrich the DNA. Add 600 µL of washing buffer PW into the CB3 column and spin at 12,000 rpm for 30 seconds to purify the DNA. Finally, add 50 µL of distilled water and spin at 12,000 rpm at room temperature for 2 minutes to elute the DNA.
11.Provide details about the controls used in dual luciferase assay.
A: Thank you for your excellent suggestion. The controls used in dual luciferase assay were given in Line396~400: A combination of 10 μg of the TQ379-proMCT reporter plasmid and the CRISPR/Act3.0-MCT-sgRNA effector plasmid was introduced into 100 μL of the protoplast suspension and gently mixed. As a control, a combination of 10 μg of the TQ379-proMCT reporter plasmid and the CRISPR/Act3.0 effector plasmid without the sgRNA was introduced into 100 μL of the protoplast suspension.
12.The entire manuscript should be revised by a native English language expert.
A: Thank you for your great suggestion. We polished the language with the assistance of an anonymous native English speaker.

Round 2
Reviewer 1 Report
Comments and Suggestions for Authors
The authors applied the changes and comments presented in the last review appropriately.